# Effects of Pretreatment Methods on Volatile Compounds in Fermented Cabernet Sauvignon Musalais by Gas Chromatography–Ion Mobility Spectrometry (GC-IMS)

**DOI:** 10.3390/foods14183172

**Published:** 2025-09-11

**Authors:** Jianlin Zhang, Buhailiqiemu Abudureheman, Minqiang Guo, Lin Chen, Qian Li, Zhuanzhuan Lv, Tiantian Long, Shuai Zhu, Haibo Pan, Xingqian Ye

**Affiliations:** 1Center for Experimental Instruction in Food Safety and Nutrition, Xinjiang Institute of Technology, Aksu 843000, China; zjl950702@163.com (J.Z.); gmqyx123@163.com (M.G.); nei3677585@163.com (L.C.); 18599340828@163.com (Q.L.); 18409496308@163.com (Z.L.); ltt1016@icloud.com (T.L.); z6988752@163.com (S.Z.); 2Key Laboratory for Quality Testing of Musalais, Aksu 843000, China; 3College of Biosystems Engineering and Food Science, Zhejiang University, Hangzhou 310058, China; apanhaibo@126.com

**Keywords:** Cabernet Sauvignon Musalais, volatile compositions, correlation, GC-IMS

## Abstract

This study investigated the correlation between flavor compounds and basic physicochemical properties of ordinary wine and Musalais wine with different pretreatment processes derived from Cabernet Sauvignon grapes. Key findings revealed significant differences in volatile compositions between Cabernet Sauvignon Musalais and the control group (ordinary wine, conventional Cabernet Sauvignon wine). Musalais samples exhibited certain commonalities in volatile profiles across different fermentation temperatures, and significant distinctions were also observed. Gas chromatography–ion mobility spectrometry (GC-IMS) identified 32 volatile compounds, with esters being the predominant contributors. Principal component analysis (PCA) and partial least squares-discriminant analysis (PLS-DA) based on the peak volumes of the 32 variables demonstrated that the first two principal components accounted for 72.2% of the cumulative variance, enabling effective differentiation of wine samples from the ordinary wine and Musalais of different treatment groups. PLS-DA further screened 13 characteristic biomarkers from the 32 volatile compounds through variable importance in projection (VIP) scoring. A correlation analysis was conducted on the basic indicators and volatile components of ordinary wine and three Musalais wines under different treatments, revealing differences both among the basic indicators (such as residual sugar, total phenolics, total acidity, total flavonoids, pH, and ethanol content) and between these indicators and volatile components. This methodology provides novel insights into crafting Cabernet Sauvignon Musalais with a unique flavor profile, advancing the optimization of fermentation strategies for traditional Chinese fruit wines.

## 1. Introduction

The traditional Chinese grape-fermented wine Musalais has a history of more than 1000 years. It is mainly produced in the northwestern part of the Taklimakan desert and the southern region of the Tianshan mountains, with Awati county being the area where its production is most concentrated. It has a unique flavor without astringency and offers health benefits such as anti-aging and nourishing properties [1] (Cheng et al., 2024). As an important part of local traditional culture, Musalais is deeply accepted by people [2] (Zhao, 2022). The primary process is to use the local “Hetian red” grape variety as the main raw material, which is made into an alcoholic beverage through crushing, pressing, boiling, and natural fermentation [3] (Zhu, 2024). The unique caramel aroma produced during the production of Musalais differs from any other wine [4] (Zhu et al., 2012). Therefore, for the same grape variety, there may be significant differences in flavor due to different processing methods. The quality differences of Musalais wine are mainly attributed to the diversity of production techniques rather than variations caused by typical grape varieties [5] (Li et al., 2024). For example, factors such as boiling time, boiling temperature, and fermentation vessels used during the Musalais wine production may lead to significant differences in the aroma produced by fermentation [6,7] (Zhu et al., 2010; Tang et al., 2025). Therefore, the ripening conditions of Musalais are more difficult to predict without precise control of temperature, environmental humidity, and even containers. The typical aroma of ordinary wine depends largely on the grape variety rather than the winemaking techniques [8,9] (Zhu et al., 2013; Zou, 2021). Accordingly, the aroma characteristics of Musalais depend more on techniques than on grape varieties. Therefore, pre-treating the Cabernet Sauvignon variety—traditionally used for red wine production—using the Musalais process, and subsequently fermenting Cabernet Sauvignon Musalais (CSM), is expected to contribute to the study of flavor profiles in different varieties of Musalais.

Understanding the types and contents of volatile compounds produced during the fermentation process helps in producing Musalais with better aroma and taste. Most aromatic compounds in Musalais are generated during the natural fermentation stage [10,11] (Zhu et al., 2019; Xu et al., 2025). Therefore, understanding the aroma evolution of Cabernet Sauvignon Musalais (CSM) is essential for controlling the quality of the finished wine, particularly for improving and stabilizing key active odor compounds and controlling off-flavors to maximize its quality [12,13] (Zhu et al., 2021; Li et al., 2024). Previous studies have explored the effects of fermentation temperature on the volatile compound profile in red wines [14,15] (Buhailiqiemu et al., 2025; Izquierdo-Cañas et al., 2020). However, there is currently no report about the impact of pretreatment methods on volatile components in CSM.

This study analyzed the changes in basic physicochemical indices (residual sugar, total phenolic, total acidity, total flavonoid, pH, and ethanol content) of CSM under different pretreatments. Volatile compounds produced by CSM under different Brix of concentrated grape juice and different fermentation temperatures were detected by gas chromatography–ion mobility spectrometry (GC-IMS). Meanwhile, principal component analysis (PCA) and partial least squares discriminant analysis (PLS-DA) were applied to identify volatile compounds that make significant contributions to the flavor profile of CSM, and correlation analysis was employed to reveal the correlations between basic physicochemical indices and volatile components. This study aims to explore the relationship between changes in volatile flavor substances and basic physicochemical indices under different pretreatments and provide new development approaches and industrial value for the Musalais industry.

## 2. Materials and Methods

### 2.1. Cabernet Sauvignon Grape

The grape variety used, Cabernet Sauvignon, was planted by Wensu Haoyuan Winery Co., Ltd., in Aksu, China. The grapes reached optimal maturity (with a sugar content of 22 °Brix) by mid-October 2024 and were manually harvested under sound sanitary conditions. The grape variety used for brewing the two different types of fruit wine in this study is Cabernet Sauvignon, and a total of 300 kg of Cabernet Sauvignon grapes were collected.

### 2.2. Wine Making Process

The brewing process of Cabernet Sauvignon wine (CSW): The grapes were divided into 4 batches. One batch was used to make wine, and the other 3 batches were used to make Musalais by the traditional method.

#### 2.2.1. Processing Technology of CSW

CSW was brewed in a 10 L glass fermentation tank on the Musalais Production Center of Xinjiang Institute of Technology. After destemming and crushing the grapes with a crusher, the crushed grapes were mixed evenly with their juice and pomace, and 100 mg/L SO_2_ was added and stirred again (Figure 1). The resulting pomace–liquid mixture was kept at 12 ± 2 °C for 24 h, then 0.6 g/L pectinase was added to the mixed grape juice for 3 h, followed by inoculation with wine yeast (Angel wine fermentation yeast powder) for alcoholic fermentation [16,17] (Li, 2023; Li et al., 2023). The fermentation temperature was 20–24 °C, lasting about one week. Subsequently, after being naturally clarified, the supernatant was filtered through eight layers of gauze, and the filtrate was further filtered through a 0.45 µm membrane filter (Millipore, Bedford, MA, USA), bottled, pasteurized at 62 °C for 25 min, stored in the dark at 10 °C, and analyzed after about one month. All Cabernet Sauvignon wines were brewed in triplicate.

#### 2.2.2. Processing Technology of Cabernet Sauvignon Musalais (CSM)

CSM was brewed in a 10 L glass fermentation tank on the Musalais Center production line of the Xinjiang Institute of Technology (Figure 1). The selected grape clusters were placed in a stainless steel tank and gently rinsed with water 2–3 times, with particular emphasis on removing dust and insect eggs from the stems. They were then transferred to a destemmer to remove the stems. The berries were crushed in a stainless steel crusher, and the juice was collected. The juice was subsequently concentrated in a stainless steel pot at high temperature (100 °C) until the sugar content reached 29 °Brix. After cooling to room temperature, the concentrated grape juice was fermented at 22 °C, 25 °C, and 28 °C, respectively. CSM fermented at the above three temperatures was simply referred to as CSM-A, CSM-B, and CSM-C. All experiments under fermentation temperature conditions were repeated three times using 10 L glass fermentation tanks. When the fermentation alcohol content was 8–14 %Vol, the fermentation was terminated. The CSM was filtered and bottled, pasteurized at 62 °C for 25 min, and then stored at a constant temperature of 10 °C in the dark.

### 2.3. Conventional Oenological Physicochemical Index Detection

The alcohol content in CSW and CSM was measured with a handheld alcohol meter. The determination of titratable acidity was based on the Chinese National Standard GB/T 15038-2006 (General Analytical Methods) [18]. According to the principle of acid-base titration, phenolphthalein was used as an indicator, and the sample was titrated with a standard alkali solution; the residual sugar content was determined by the anthrone-sulfuric acid colorimetric method; and the pH value was measured using a pH meter (Mettler-Toledo, Schwerzenbach, Switzerland).

The total phenol content was determined by the Folin–Ciocalteu colorimetric method [19] (Cao et al., 2023). After the reaction, the generated gallic acid was scanned at full wavelength in the range of 400 nm to 900 nm, and the maximum absorption peak was located at 765 nm. The absorbance value was measured at 765 nm to calculate the total phenol content. The calculation formula is shown in Equation (1).(1)$$Total\;phenol\;content=\frac{VN(A-0.0215)}{3.7366m}$$

V: total volume of the extract (mL); N: dilution factor; A: measured absorbance; m: sample mass

The determination of the total flavonoid content was carried out with reference to the NaNO_2_-Al(NO_3_)_3_ color development method in the literature [20,21] (Zununa et al., 2022; Osorio Alises et al., 2024). Specifically, 5 mg of rutin was accurately weighed, dissolved in 60% absolute ethanol, and transferred to a 25 mL volumetric flask, which was then filled to the mark with the same solvent. Pipettes (0, 1.0, 1.5, 2.0, 2.5, 3.0, and 3.5 mL) of the rutin solution were separated into 10 mL volumetric flasks. Then, 0.3 mL of 5% NaNO_2_ solution was added to each flask, followed by shaking and standing. After that, 0.3 mL of 10% Al(NO_3_)_3_ solution was added, shaken again, and left to stand. Subsequently, 4 mL of 4% NaOH solution was added, and the mixture was shaken. A small amount of 60% absolute ethanol was used to fix the volume to the mark, and after standing, the absorbance value was measured at 505 nm. The total flavonoid content was calculated using Equation (2).(2)$$Total\;flavonoid\;content\;(mg/g)=\frac{XV}{M}$$

X: the mass concentration of total flavonoids in wine samples (mg/mL); V: original extract volume (mL); M: sample mass (g)

### 2.4. Gas Chromatography–Ion Mobility Spectrometry (GC-IMS)

Gas chromatography–ion mobility spectrometry (GC-IMS) analysis was performed by using a GC-IMS instrument (FlavourSpec^®^, G.A.S GmbH, Dortmund, Germany) equipped with an MXT-WAX capillary column (30 m × 0.53 mm × 1 µm, Restek, USA), with reference to the experimental methods of Jiang and Jiang [22,23] (Jiang et al., 2024; Jiang et al., 2024) with slight modifications. The preparation method for each Musalais sample was as follows: 0.5 mL of the sample was added to a 20 mL headspace vial, sealed with a magnetic cap, and incubated at 60 °C for 25 min. Subsequently, 200 µL of headspace gas was injected into the inlet at a flow rate of 60 mL/min using a syringe preheated to 85 °C. The column temperature was set at 60 °C, and the drift tube was maintained at 45 °C. The drift gas flow rate was set at 150 mL/min. High-purity nitrogen (purity 99.99%) was used as the carrier gas, and the carrier gas flow rate through the GC column was programmed as follows: 2 mL/min for the first 2 min, 10 mL/min for the next 10 min, 50 mL/min for the subsequent 15 min, 100 mL/min for the following 20 min, and 150 mL/min for the final 30 min. The retention indices (RI) of volatile organic compounds (VOCs) were calculated using n-butanone to n-nonanone (C4–C9) as references. VOCs were identified by comparing the ion drift time and retention time with those of standard substances in the GC-IMS library. Each sample was analyzed once, and each VOC was relatively quantified based on the peak area. The standard solutions of n-butanone to n-nonanone (C4–C9) were purchased from Sinopharm Chemical Reagent Co., Ltd. (Shanghai, China).

### 2.5. Multivariate Statistical Analysis

Multivariate statistical methods are capable of handling and analyzing relationships between multiple variables (features), providing strong support for volatile flavor analysis, classification, and screening through approaches such as classification, dimensionality reduction, and regression modeling applied to multiple independent variables [24,25] (Liao et al., 2023; Xu, 2021). Among them, correlation analysis is a statistical method to study the correlation between random variables. Partial least squares (PLS) extracts typical components of two groups of variables by extracting principal components [26] (Li et al., 2021). Cluster analysis, an unsupervised learning method, analyzes datasets to understand the internal connections of data, with comprehensive and scientific results [27] (Yuan et al., 2023). Principal component analysis (PCA), a dimensionality reduction method in multivariate statistics, recombines original multiple variables into a set of uncorrelated comprehensive variables through linear transformation, retaining as much information as possible from the original variables and reducing them into new variables or principal components. It is widely applied in the food industry and has proven feasibility [28,29] (Qi et al., 2024; Gao et al., 2022).

### 2.6. Statistical Analysis

The volatile substances in the samples were qualitatively analyzed using the GC×IMS Library Search NIST database and IMS database built into the VOCal 0.4.03 software. The peporter plugin was used to view the two-dimensional spectra of the samples and analyze the differences between the spectra. The gallery plot plugin was employed to draw fingerprint spectra. Origin 2022 was used for drawing bar charts and pie charts. SIMCA 14.1 software was used for principal component analysis (PCA) and partial least squares discriminant analysis. Pearson correlation analysis was used to determine the correlation, which was plotted through the website https://www.chiplot.online/, accessed on 25 May 2025.

## 3. Results and Discussion

### 3.1. Physicochemical Index Analysis

The total acid and flavonoid contents in CSM-A, CSM-B, and CSM-C were higher than those in the CSW group (*p* < 0.05), while the total phenol content of CSM-A, CSM-B, and CSM-C was lower than that in the CSW group (Figure 2). This indicates that different pretreatment methods significantly affected the composition of CSW and CSM. Since CSM undergoes natural fermentation after concentration of boiling and cooling, the polyphenol content is partially destroyed [30] (Qu et al., 2020), making it much lower than that in CSW. In contrast, the total acid content in CSM was higher than that in CSW due to the longer fermentation cycle. Interestingly, the residual sugar content of CSM was much higher than that of CSW. This was mainly due to the increase in sugar content during the boiling, concentration, and the natural fermentation process, where the activity of yeast was weaker than that of commercial yeast. The total phenol content of CSM was much lower than that of CSW. In contrast, flavonoids, alcohol content, and pH among CSW showed small overall differences. On the other hand, there were significant differences in all indices except total phenol under different temperature treatments among the CSM-A, CSM-B, and CSM-C groups, while total phenol showed no differences among different temperature treatment groups. This suggested that boiling had a significant effect on total phenol. At lower fermentation temperatures, it showed the characteristics of slow fermentation, weak microbial activity, more total acid accumulation, and better flavonoid content, but the lower fermentation temperatures caused incomplete fermentation, resulting in low alcohol content, high residual sugar, and relatively stable pH. The increase in temperature accelerated the fermentation rate, caused vigorous microbial metabolism, low organic acids, increased alcohol content, and decreased residual sugar, pH, total acid, and flavonoid content due to oxidation and other reactions [31,32] (Zhang et al., 2018; Cecile et al., 2022). However, if the temperature is too high, the activity of microorganisms will be inhibited or even inactivated, leading to abnormal fermentation. As a result, the increase in alcohol content will be limited, and the residual sugar will increase. Meanwhile, intensified decomposition of substances caused by high temperatures results in significant loss of flavonoids, thereby affecting the quality of Musalais. Overall, different pretreatment methods for brewing the same grape variety lead to significant differences in quality.

### 3.2. Identification of Volatile Compounds

The volatile components of CSW and CSM-A, CSM-B, and CSM-C were analyzed by GC-IMS. Figure 3A shows the three-dimensional topographic maps of the GC-IMS spectra of volatile organic compounds in CSW and CSM-A, CSM-B, and CSM-C. The *X*-axis represents retention time, the *Y*-axis represents drift time, and the *Z*-axis represents ion peaks. Each point in the spectrum represents a volatile organic compound. The shade of color indicates the concentration level, where white represents low concentration, red represents high concentration, and the darker the color, the higher the concentration. Depending on the concentration and properties of the volatile components, the compound may produce one, two, or more spots (representing monomers, dimers, or trimers) [33,34] (Zhang et al., 2023; Miao et al., 2023). The three-dimensional topographic maps of the CSW and CSM-A, CSM-B, and CSM-C samples obtained by the LAV analysis are similar, making it difficult to distinguish volatile compounds in detail (Figure 3A). The three-dimensional topographic map of GC-IMS was projected onto a two-dimensional plane vertical view for the CSW and CSM-A, CSM-B, and CSM-C samples to obtain their two-dimensional topographic maps (Figure 3B). To more intuitively reflect the differences in volatile components between different groups, comparative spectra of the samples were plotted (Figure 3C). The specific spectral information of CSW and CSM-A, CSM-B, and CSM-C under different pretreatment groups varies, which may be attributed to the differences in specific volatile components among different process treatment groups. Particularly, there are differences in volatile components between the CSW group and the CSM-A, CSM-B, and CSM-C groups under different process treatments. Among CSM-A, CSM-B, and CSM-C under the same process conditions but different fermentation temperatures, CSM-B exhibited a relatively higher concentration.

### 3.3. Qualitative Analysis of Flavor Components

The VOCal 0.4.03 software of the instrument was used to conduct a qualitative analysis of the volatile compounds in the CSW group and CSM-A, CSM-B, and CSM-C processed by different treatments. The results of the qualitative analysis of volatile compounds are shown in Figure 4A. As listed in Table 1, a total of 32 volatile compounds were tentatively identified by GC-IMS, including five aldehydes, seven ketones, three alcohols, 12 esters, one acid, one ether, one alkene, and one other compound (aldehydes accounted for 15.6%; ketones for 21.9%; alcohols for 9.4%; esters for 37.5%; and acids, ethers, benzenes, alkenes, and others each accounted for 3.1%) (Figure 4B). Combining with Table 1, esters such as ethyl butyrate and methyl propionate were more abundant in variety and quantity than other volatile compounds, serving as the main sources of the fruit wine’s aroma and thereby presenting a predominantly fruity fragrance [35] (Li et al., 2023), followed by ketones such as 3-hydroxy-2-butanone and 3-octanone, which were auxiliary aroma substances contributing to the caramel aroma of Musalais wine [36] (Chen., 2022). The interaction of different types of volatile aroma substances contributes to the rich, mellow, smooth, and delicate style characteristics of Musalais wine [37] (Zhang et al., 2018).

As shown in the percentage accumulation diagram (Figure 4C), significant differences were observed in the volatile components among CSW and CSM-A, CSM-B, and CSM-C under different pretreatment conditions. Compared with CSM, CSW did not detect four categories: acids, aromatic hydrocarbons, alkenes, and others. This may be attributed to raw materials and technological processing methods. In addition, the contents of aldehydes, alcohols, and ethers in the CSW group were higher than those in CSM-A, CSM-B, and CSM-C; the ketone content of CSW was lower than that in CSM-A, CSM-B, and CSM-C. For CSM-A, CSM-B, and CSM-C, the volatile components also showed variations due to different fermentation temperatures. With the increase in fermentation temperature, the contents of aldehydes, acids, and alcohol gradually decreased, while the ester content gradually increased. Meanwhile, the pleasant aromas of esters, such as fruity, floral, and sweet notes, have a significant impact on the final flavor of Musalais.

The data in Table 1 were visualized using Venn diagram analysis (Figure 4D). It can be seen that there are 32 volatile components, among which CSW contains 16, CSM-A contains 30, CSM-B contains 28, and CSM-C contains 20. CSM-A has more volatile components, while CSW has the most single volatile components, indicating that volatile components are greatly influenced by different pretreatments. CSW has no common components with CSM-A, CSM-B, and CSM-C, indicating that different pretreatments have great differences in the flavor of CSW. Under the same treatment but different temperature treatments, CSM-A, CSM-B, and CSM-C share 15 volatile components, accounting for 46.9% of the total components, indicating that the three Musalais under the same treatment but different fermentation temperatures have commonalities in volatile components. With the increase in fermentation temperature, the contents of esters and acids in Musalais wine decrease, and under the same process but different fermentation temperatures, the aroma complexity of the wine decreases significantly with temperature elevation [38] (Zhang et al., 2023).

### 3.4. Fingerprint Spectrum Analysis of CSW and CSM-A, CSM-B, CSM-C

To further compare the differences in volatile substances among CSW and CSM-A, CSM-B, and CSM-C, fingerprint spectra were automatically generated using the signal peaks of various substances from the GC-IMS two-dimensional spectra to identify the characteristic peak regions of different fruit wine samples (Figure 4E). The horizontal axis of the spectrum represents different volatile flavor substances detected by GC-IMS, and the vertical axis shows the CSW sample and CSM-A, CSM-B, and CSM-C samples in different treatment groups. Each horizontal row indicates all selected signal peaks of samples at that level, and each column indicates the content variation of a substance across different samples. The brightness of a single point represents the content level of a volatile substance, where a darker brightness indicates a higher content of the substance [39] (Li et al., 2024).

There were differences between CSW and CSM-A, CSM-B, and CSM-C (concentrated in area a). The overall content of volatile substances in CSW was lower than that in CSM, but the contents of (E,E)-2,4-Hexadienal, methyl acetate, 3-Octanol, (Z)-4-heptenal, 2-Methyl-2-pentenal, and hydroxyacetone were significantly higher than those in CSM. These volatile components in CSM-A, CSM-B, and CSM-C were basically consistent (concentrated in areas c and d). Area c had higher contents of volatile components, including 2-methylpropanol-M, camphene, 3-mercapto-2-butanone, etc. These components endowed Musalais wine with flavors of jasmine, apple, and rose. In particular, 3-mercapto-2-butanone gave Musalais wine a special caramel aroma, making the wine’s aroma fuller and more prominent. Area d had lower contents of volatile components, including 2-Methyl-2-pentenal, hydroxyacetone, (Z)-4-heptena, 3-octanol, methyl acetate, and (E,E)-2,4-Hexadienal, which moderately increased the green, fruity, and caramel flavors of Musalais wine. Overall, with the increase in fermentation temperature, the contents of 2-methylpropanol-M, dimethyl disulfide, and isopropyl butanoate increased, while the contents of butanoic acid ethyl ester-D, 2-furanmethanol,5-methyl, 2-butanol, and 2-methyl-2-pentenal decreased, indicating significant differences in the volatile contents of CSM with fermentation temperature changes. Higher fermentation temperatures result in more obvious pungent odors (such as dimethyl disulfide), while lower temperatures increase the complexity of the fruit wine’s flavor, making the flavor of CSM the richest.

### 3.5. Multivariate Statistical Analysis of Volatile Substances

Based on the differences in volatile substance contents, the peak volumes of 32 characteristic regions in the fingerprint spectra were used as characteristic variables for PCA analysis of the four treatment groups. The contribution rate of the first principal component, PC1, was 23.2%, and that of the second principal component, PC2, was 49.0%, with a cumulative contribution rate of 72.2% for the first two principal components (Figure 5A). This indicates that PC1 and PC2 collectively account for most of the information from the original variables and can represent the main characteristics of volatile flavors in different treatment groups [40] (Shao et al., 2023). After PCA transformation of the wine sample data from different treatment groups, the four groups of pretreated fruit wine samples were clustered separately, except for CSM-A and CSM-B, suggesting certain differences between CSM samples under different process treatments and different fermentation temperatures within the same process. This is consistent with the results of GC-IMS fingerprint spectrum analysis. Cluster analysis of VOCs in CSW under different treatment groups revealed distinct grouping patterns (Figure 5E). The color scale from blue to yellow indicates the change in relative substance concentration from low to high. Key volatile substances corresponding to CSW and CSM under different treatment groups can be clearly distinguished. The cluster analysis results are consistent with the PCA results, indicating that different process pretreatments and different fermentation temperatures within the same process significantly affect the formation of VOCs in CSW and CSM.

To further screen characteristic markers and evaluate the contribution of different volatile substances to various wine samples, we used the PLS-DA statistical method. Variable data and classification information were divided into training and testing sets to establish a classification model, and differences between groups were identified with the help of grouping information through projection and discriminant analysis [41] (Wang et al., 2025). The PLS-DA score plot was drawn using the 32 identified volatile aroma substances in the four groups of samples as dependent variables, and different treatment groups as independent variables (Figure 5B). The plot showed the same trend as the PCA analysis, with more concentrated samples between CSW and CSM. By randomly changing the order of classification variables to establish corresponding models, R^2^X and R^2^Y represent the percentages of information in the X and Y matrices explained by the PLS-DA classification model, respectively. The closer their values are to 1, the better the model performance, and the difference between them should not be too large [42] (Jin et al., 2023). Q^2^, calculated through cross-validation, evaluates the predictive ability of the PLS-DA model, where a larger Q^2^ indicates better predictive performance. The performance index results of the model obtained through 200 sequential permutation tests in PLS-DA (Figure 5C): the independent variable fitting index R^2^X is 0.986, the dependent variable fitting index R^2^Y is 0.996, |R^2^X − R^2^Y| < 0.15, and the model prediction index Q^2^ is 0.949. Additionally, the intercept point of the Q^2^ regression line with the vertical axis was less than 0, indicating no overfitting in the model and validating its effectiveness for aroma discrimination analysis of the four groups of wine samples.

In the PLS-DA model, the Variable Importance for the Projection (VIP) value was used to further distinguish the contribution of different aroma substances to wine samples in different treatment groups [43] (Zhang et al., 2023). The horizontal axis of Figure 5D corresponds to the numbered volatile substances in Table 1, and the vertical axis represents the VIP value of each substance. According to the criterion of VIP > 1, 13 characteristic marker substances were screened out, including seven esters, one ketone, one alcohol, one alkene, one acid, one ether, and one heterocycle. These substances are (1) esters: methyl propanoate, butanoic acid ethyl ester-M (monomer), ethyl hexanoate, 2-Methylbutyl acetate, isopropyl butanoate, acetic acid ethyl ester-M (monomer), and ethyl 3-methylbutanoate (isoamyl valerate); (2) ketone: 3-mercapto-2-butanone; (3) alcohol: 2-butanol; (4) alkene: camphene; (5) acid: crotonic acid; (6) ether: 2-pentyl furan; and (7) heterocycle: dimethyl disulfide. These key substances significantly influence the differentiation of volatile profiles among treatment groups. For example, esters (e.g., methyl propanoate and ethyl hexanoate) contribute fruity and floral aromas, while 3-mercapto-2-butanone is associated with caramel-like notes. Dimethyl disulfide may impart pungent or sulfurous characteristics, and camphene adds terpenoid-like freshness. The screening of these markers provides a scientific basis for distinguishing the flavor characteristics of CSW and CSM under different processes and fermentation temperatures.

### 3.6. Correlation Analysis Between Physicochemical Indices and Volatile Substances

Blue indicates a positive correlation, red indicates a negative correlation, and points represent the correlation, with larger points indicating stronger correlation (Figure 6) [23] (Jiang et al., 2024). Alcohol content was positively correlated with Ethyl 3-methylbutanoate and Isophorone, and negatively correlated with total flavonoid 2-Furanmethanol, 5-methyl, but showed weak correlation with other volatile compounds. Residual sugar was significantly positively correlated with 2-Methylbutyl acetate, Benzaldehyde, butylbenzene, 2-Methylpropanol-D, Acetic acid ethyl ester-M, total acidity, and pH, while significantly negatively correlated with 2-Butanone, 3-hydroxy, 3-Octanol, (Z)-4-heptenal, propyl bytanoate, dimethyl disulfide, methyl acetate, pH, and total phenolic. Total phenolic was significantly positively correlated with 3-Octanol, (Z)-4-heptenal, 2-Butanone, 3-hydroxy, butanoic acid ethyl ester-D, methyl propanoate, methyl acetate, and 2-Methyl-2-pentenal, and significantly negatively correlated with benzaldehyde, butylbenzene, 3-octanone, ethyl hexanoate, 3-mercapto-2-butanone, and 2-pentylfuran. Total acidity was significantly positively correlated with propyl bytanoate, 2-Furanmethanol, 5-methyl, and hydroxyacetone, and significantly negatively correlated with 2-Methylbutyl acetate, 3-octanone, 3-mercapto-2-butanone, 2-butanol, and acetic acid ethyl ester-D. pH was significantly positively correlated with 2-Butanone, 3-hydroxy, 3-Octanol, (Z)-4-heptenal, and ethyl 3-methylbutanoate, and significantly negatively correlated with benzaldehyde, butylbenzene, butanoic acid, 3-methyl-butyl ester, and isopropyl butanoate. Total acidity was significantly negatively correlated with 2-Butanone, 3-hydroxy, 3-Octanol, (Z)-4-heptenal, methyl propanoate, and 2-Methyl-2-pentenal. Residual sugar and total acidity were positively correlated with 3-mercapto-2-butanone, thereby promoting the rapid formation of caramel flavor. Additionally, different basic indices such as residual sugar and total acidity exhibited distinct effects on the flavor of CSW. Combined with Figure 2, CSM under different processes and fermentation temperatures showed differences in basic physicochemical indices, indicating that the formation of flavor substances is closely related to the different treatment methods of CSM.

## 4. Conclusions

The volatile substances in CSW under different treatment groups were qualitatively analyzed by GC-IMS combined with chemometric methods. The results showed that a total of 32 known volatile substances were detected through qualitative analysis of flavor compounds. Based on peak volume, PCA analysis was performed, and 13 volatile substances with a VIP > 1 were screened out by the PLS-DA method, including methyl propanoate, butanoic acid ethyl ester-M (monomer), ethyl hexanoate, 2-Methylbutyl acetate, 2-butanol, isopropyl butanoate, etc. These components endow CSW with unique flavors, which can be enhanced by applying research-based winemaking techniques. Through cluster analysis, heatmaps, and correlation analysis, it was found that the volatile components of the CSW group differed significantly from those of the CSM-A, CSM-B, and CSM-C groups. In the CSW group, the relative contents of (E,E)-2,4-Hexadienal, methyl acetate, 3-Octanol, (Z)-4-heptenal, and 2-Methyl-2-pentenal were higher. Under the same treatment but different fermentation temperatures, the differences among the CSM-A, CSM-B, and CSM-C groups were smaller, with components such as 2-methylpropanol-M, camphene, and 3-mercapto-2-butanone showing higher concentrations. These components impart flavors of jasmine, apple, and rose to CSM, particularly 3-mercapto-2-butanone, which gives it a unique caramel aroma. Optimizing grape varieties and process parameters can further enhance these aroma profiles, thereby providing a theoretical basis and potential directions for the future development of Musalais fruit wine.

## Figures and Tables

**Figure 1 foods-14-03172-f001:**
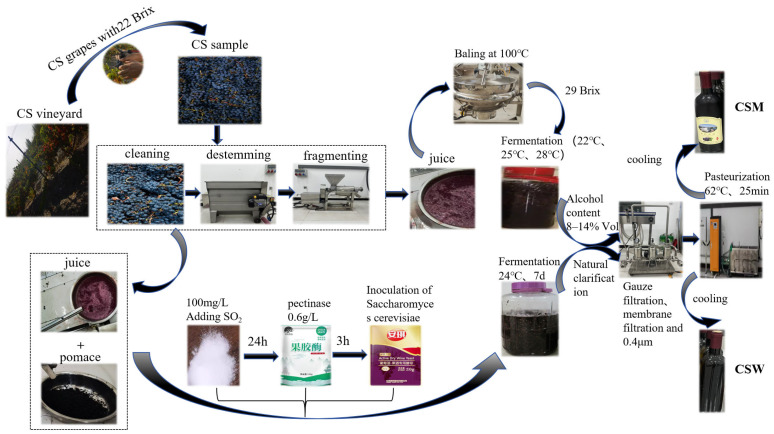
Brewing processes of Cabernet Sauvignon wine and Musalais.

**Figure 2 foods-14-03172-f002:**
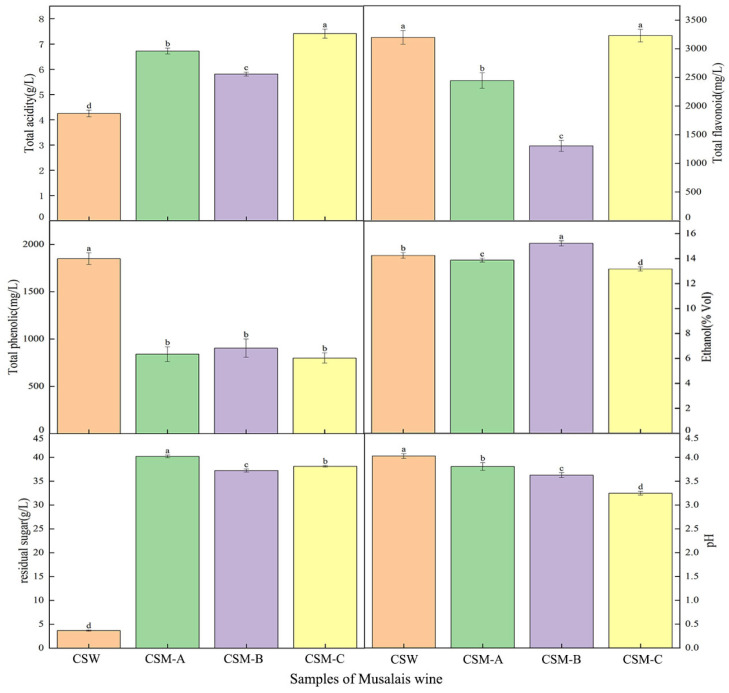
Basic physicochemical indices of Cabernet Sauvignon wine and Cabernet Sauvignon Musalais under different pretreatments. Notes: Cabernet Sauvignon wine (CSW) fermented at 28 °C; Cabernet Sauvignon Musalais (CSM-A, CSM-B, and CSM-C) fermented at 22 °C, 25 °C, and 28 °C, respectively. Different lowercase letters indicate that there are statistically significant differences in total acidity, residual sugar content, pH value, ethanol, total phenol content, and total flavonoid content among different treatment groups.

**Figure 3 foods-14-03172-f003:**
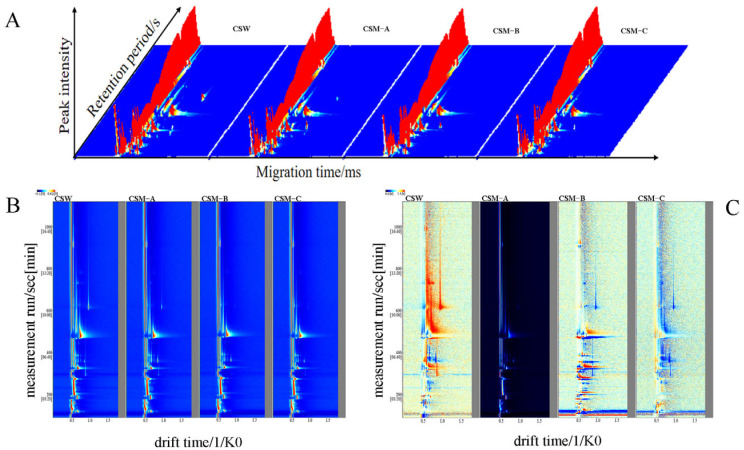
Three-dimensional GC-IMS spectra (**A**), two-dimensional GC-IMS spectra (**B**), and differential GC-IMS spectra (**C**) of volatile substances in CSW group and CSM-A, CSM-B, and CSM-C samples under different pretreatments.

**Figure 4 foods-14-03172-f004:**
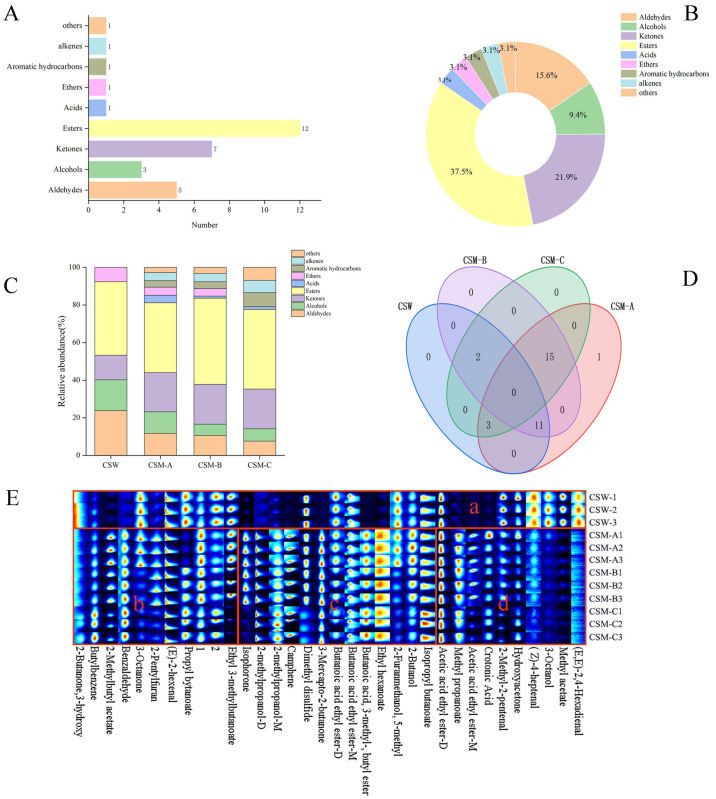
Characteristics of volatile compounds in CSW and CSM-A, CSM-B, CSM-C: number (**A**) and percentage (**B**) of volatile compounds, proportion of volatile compounds (**C**), Venn diagram (**D**), and fingerprint spectrum (**E**).

**Figure 5 foods-14-03172-f005:**
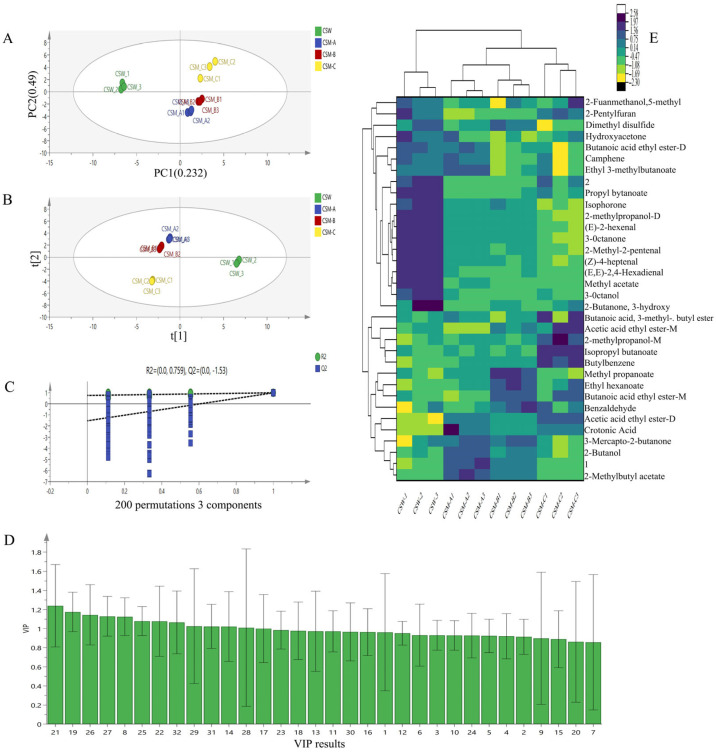
PCA analysis (**A**), PLS-DA score plot (**B**), confidence test results (**C**), VIP value distribution of volatile substances (**D**), and cluster heatmap analysis (**E**).

**Figure 6 foods-14-03172-f006:**
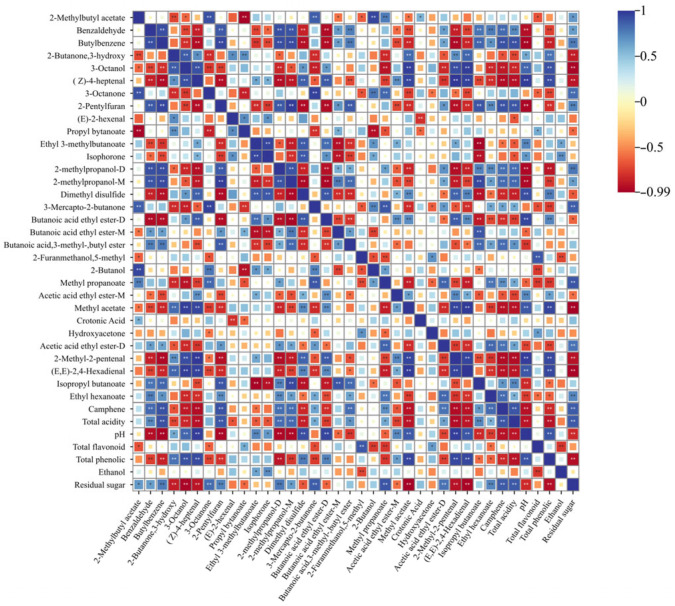
Correlation analysis between physicochemical indices and volatile substances. Note: “*” indicates a significant difference; “**” indicates a highly significant difference.

**Table 1 foods-14-03172-t001:** Volatile compounds of CSW and CSM-A, CSM-B, CSM-C samples.

Serial Number	Volatile Compounds	Aroma Description	CAS Number	Molecular Formula	RI	RT/s	Dt/ms
Aldehydes							
1	Benzaldehyde	almond-like, sweet, caramel-like	C100527	C_7_H_6_O	1474.9	919.358	1.14738
2	(Z)-4-heptenal	green aroma, fruity aroma	C6728310	C_7_H_12_O	1237.9	518.674	1.62804
3	(E)-2-hexen1al	grassy aroma, apple	C6728263	C_6_H_10_O	1219	486.481	1.51961
4	2-Methyl-2-pentenal	green aroma, fruity aroma, floral aroma	C623369	C_6_H_10_O	841.3	158.422	1.15562
5	(E,E)-2,4-Hexadienal	green aroma, fruity aroma, floral aroma	C142836	C_6_H_8_O	906.1	187.469	1.46269
Alcohols							
6	3-Octanol	oil, nut, herbal aroma	C589980	C_8_H_18_O	1367.9	733.897	1.39571
7	2-Furanmethanol, 5-methyl	caramel, bread-like aroma	C3857258	C_6_H_8_O_2_	971	221.859	1.2639
8	2- butanol	fruity aroma	C78922	C_4_H_10_O	1022.7	255.915	1.33845
Ketones							
9	2-Butanone, 3-hydroxy	creaminess, strawberry	C513860	C_4_H_8_O_2_	1295.6	630.271	1.32738
10	3-Octanone	creaminess, rose, jasmine	C106683	C_8_H_16_O	1236.3	515.776	1.31371
11	Isophorone	minty aroma, woody aroma	C78591	C_9_H_14_O	1114.7	338.326	1.25032
12	2-Methylpropanol-D	subtle apple aroma	C78831	C_4_H_10_O	1105.3	327.345	1.3761
13	2-Methylpropanol-M	jasmine aroma, apple aroma, rose aroma	C78831	C_4_H_10_O	1111.3	334.333	1.172
14	3-mercapto-2-butanone	caramelized aroma and grilled meat aroma	C40789988	C_4_H_8_OS	1269.3	576.985	1.12669
15	Hydroxyacetone	creaminess, caramel aroma	C116096	C_3_H_6_O_2_	718	115.018	1.2284
Esters							
16	Propyl bytanoate	banana scent, pineapple scent	C105668	C_7_H_14_O_2_	1154	388.337	1.25775
17	Ethyl 3-methylbutanoate	apple aroma, banana aroma, strawberry aroma	C108645	C_7_H_14_O_2_	1099.4	320.638	1.2474
18	Butanoic acid ethyl ester-D	apple aroma, buttery aroma	C105544	C_6_H_12_O_2_	1046	273.944	1.55499
19	Butanoic acid ethyl ester-M	pineapple aroma, banana aroma	C105544	C_6_H_12_O_2_	1047.3	274.946	1.20355
20	Butanoic acid, 3-methyl-, butyl ester	banana aroma, pineapple and apple aroma	C109193	C_9_H_18_O_2_	1047.7	275.28	1.38104
21	Methyl propanoate	strawberry, pineapple	C554121	C_4_H_8_O_2_	919.6	194.147	1.09882
22	Acetic acid ethyl ester-M	Banana, grassy aroma, apple aroma	C141786	C_4_H_8_O_2_	887.2	178.455	1.09705
23	Acetic acid ethyl ester-D	apple aroma, banana aroma, grassy aroma	C141786	C_4_H_8_O_2_	903.4	186.134	1.34555
24	Methyl acetate	fruity aroma, pineapple aroma	C79209	C_3_H_6_O_2_	846.2	160.425	1.19467
25	Isopropyl butanoate	pineapple, banana	C638119	C_7_H_14_O_2_	1048.1	275.614	1.2639
26	Ethyl hexanoate	apple aroma, banana aroma	C123660	C_8_H_16_O_2_	1215.2	480.241	1.7945
27	2-Methylbutyl acetate	the aroma of fruits such as apples and pears	C624419	C_7_H_14_O_2_	1133.2	361.012	1.74368
Acids							
28	Crotonic acid	irritating smell	C107937	C_4_H_6_O_2_	842.1	158.756	1.1148
Ethers							
29	Dimethyl disulfide	irritating smell	C624920	C_2_H_6_S_2_	1098.4	319.516	1.12975
Aromatic hydrocarbons							
30	Butylbenzene	neroli, jasmine, pineapple	C104518	C_10_H_14_	1294.7	628.822	1.20666
Alkenes							
31	Camphene	pine scent, lemon fragrance	C79925	C_10_H_16_	926.6	197.7	1.21109
others							
32	2-pentyl furan	peachy aroma, almond aroma	C3777693	C_9_H_14_O	1234.2	512.152	1.24311

## Data Availability

The data presented in this study are available upon request from the corresponding author. The data are not publicly available due to privacy reasons.

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
