# Peer review of "Effects of Pretreatment Methods on Volatile Compounds in Fermented Cabernet Sauvignon Musalais by Gas Chromatography–Ion Mobility Spectrometry (GC-IMS)"

_foods, 2025, doi:10.3390/foods14183172_

Round 1
Reviewer 1 Report
Comments and Suggestions for Authors
1. The manuscript claims novelty in exploring “pretreatment methods,” but the treatments tested are essentially limited to juice concentration + three fermentation temperatures. This is rather narrow compared to broader pretreatments (enzymatic treatments, SOâ‚‚ adjustments, vessel type, etc.). It is suggested to define why these treatments are significant “pretreatments” and clarify the contribution to Musalais research beyond existing studies on fermentation temperature in Introduction section.
2. Many identifications of volatile compounds remain tentative without GC-MS confirmation. The risk of misidentification should be explicitly acknowledged in discussion. Authors should clearly state that these identifications are preliminary (“putative”) until confirmed with a more robust method such as GC-MS.
3. As the study did not conduct sensory evaluation, the claims about providing a foundation for industrial diversification of Musalais in conclusion section are overstated. Revised this sentence.
4. Figures are dense (e.g. Figure 4, Figure 5). Consider simplifying or moving supplementary details out of the main text. The resolution of figures is unclear. Please revise it.
5. Please check the grammar, article needs extensive English editing service. Check also consistency.
For example:
- Line 37–38: “It is mainly producted in the northwestern part of the Taklimakan desert …” → Typo: producted → produced.
- Line 463: “Key Laboratory of Musalles quality testing” --> (Musalais)
- Line 433–435: “… a total of 31 known volatile substances were detected …”
→ Inconsistent: elsewhere the text says 32 compounds. Needs correction for consistency.
Reviewer 2 Report
Comments and Suggestions for Authors
The study: "Effects of Pretreatment Methods on Volatile Compounds in Fermented Cabernet Sauvignon Musalais by Gas Chromatography-Ion Mobility Spectrometry (GC-IMS)" investigates a relevant topic by analyzing the impact of the Musalais production method (involving boiling concentration and different fermentation methods) on the volatile compound profile and physicochemical properties of a Cabernet Sauvignon Musalais beverage, compared to conventional wine.
After a thorough review, I have provided several recommendations below that are essential for strengthening the manuscript before it can be considered for publication.
- The manuscript requires comprehensive rewriting for conciseness and clarity. The current text often feels unnatural and does not flow smoothly in scientific English.
- Many sentences are excessively long and complex, hindering comprehension. I strongly recommend breaking them into shorter, more precise sentences.
- There is a noticeable influence of direct translation from another language, resulting in awkward word order and non-idiomatic lexical choices. A thorough revision by a native English speaker or a professional academic editing service is highly advised to ensure natural and standard scientific phrasing.
- The Musalais production method is not described with sufficient clarity for the reader to easily understand or replicate the process. This section must be expanded and explained in a more detailed, step-by-step manner.
- A significant shortcoming is the complete lack of sensory analysis. Instrumental data on volatile compounds is greatly enhanced and validated by correlating it with human perception. Incorporating a descriptive sensory analysis or consumer preference test is essential for the study to be considered complete and robust.
- To make a truly significant comparison regarding the impact of the Musalais method (vs. grape variety), the study should be expanded to include other grape varieties processed using both the conventional and Musalais techniques. Currently, the findings are limited to a single variety.
- The conclusions need to be more strongly supported by the data. The manuscript should clearly demonstrate: how the boiling process specifically leads to reduced polyphenol content while increasing acidity and residual sugar; how and why different fermentation temperatures significantly influence the aromatic profile (e.g., complex aromas at lower temperatures vs. flavonoid loss and sulfur compound formation at higher temperatures).
- Please avoid the term "volatile flavor compound". The standard and preferred term in scientific literature is "volatile compound" or "aroma compound". "Flavor" is a multimodal sensation that includes taste (sweet, sour, etc.), which these instruments do not measure.
- Ensure strict adherence to the author guidelines of the target journal.
-The formatting of references (citation style, journal abbreviations, etc.).
-The correct way to cite literature within the text.
-The structure of headings and subheadings.
-Standardize the scientific style throughout the manuscript and eliminate redundant phrasing.
Author Response
Response to reviewer 2
Comments 1: The manuscript requires comprehensive rewriting for conciseness and clarity. The current text often feels unnatural and does not flow smoothly in scientific English.
Response 1:Thank you for your suggestion. The entire manuscript has been comprehensively rewritten.
Comments 2: Many sentences are excessively long and complex, hindering comprehension. I strongly recommend breaking them into shorter, more precise sentences.
Response 2:Thank you for your suggestion.The overly lengthy and complex sentences have been revised.
Comments 3: There is a noticeable influence of direct translation from another language, resulting in awkward word order and non-idiomatic lexical choices. A thorough revision by a native English speaker or a professional academic editing service is highly advised to ensure natural and standard scientific phrasing.
Response 3:Thank you for your suggestion.The manuscript has been comprehensively revised.
Comments 4: The Musalais production method is not described with sufficient clarity for the reader to easily understand or replicate the process. This section must be expanded and explained in a more detailed, step-by-step manner.
Response 4:Thank you for your suggestion. The Musalais production process has been elaborated in detail, with specific refinements as follows: (Fig.1) The selected grape clusters were placed in a stainless steel tank and gently rinsed with water 2-3 times, with particular emphasis on removing dust and insect eggs from the stems. They were then transferred to a destemmer to remove the stems. The berries were crushed in a stainless steel crusher, and the juice was collected. The juice was subsequently concentrated in a stainless steel pot at high temperature (100°C) until the sugar content reached 29°Brix. After cooling to room temperature, the concentrated grape juice was fermented at 22 ℃, 25 ℃ and 28 ℃ respectively. The Cabernet Sauvignon Musalais fermented at the above three temperatures were simply referred to as CSM-A, CSM-B and CSM-C. All experiments under fermentation temperature conditions were repeated three times using 10-liter glass fermentation tanks. When the fermentation alcohol content was 8-14 %Vol, the fermentation was terminated. The Cabernet Sauvignon Musalais was filtered and bottled, pasteurized at 62 ℃ for 25 minutes, and then stored at a constant temperature of 10 ℃ in the dark.
Comments 5: A significant shortcoming is the complete lack of sensory analysis. Instrumental data on volatile compounds is greatly enhanced and validated by correlating it with human perception. Incorporating a descriptive sensory analysis or consumer preference test is essential for the study to be considered complete and robust.
Response 5:Thank you for your suggestion. Incorporating descriptive sensory analysis would indeed be highly valuable; however, as the flavor characteristics of Cabernet Sauvignon Musalais have not been previously reported, this study focuses primarily on comparing the flavor profiles of Cabernet Sauvignon Musalais and traditional Cabernet Sauvignon wine using GC-IMS. Consequently, sensory analysis was not conducted in the current phase. Future research will take into account the importance of sensory evaluation and consumer preference studies.
Comments 6: To make a truly significant comparison regarding the impact of the Musalais method (vs. grape variety), the study should be expanded to include other grape varieties processed using both the conventional and Musalais techniques. Currently, the findings are limited to a single variety.
Response 6:Thank you for your suggestion. Given that Cabernet Sauvignon grapes are commonly used in winemaking, while Cabernet Sauvignon Musalais has not been previously reported, our initial research focus was urgently directed toward understanding the flavor differences between Cabernet Sauvignon Musalais and traditional Cabernet Sauvignon wine, with the aim of facilitating potential production applications. In subsequent studies, comparative analyses will be extended to include other grape varieties.
Comments 7: The conclusions need to be more strongly supported by the data. The manuscript should clearly demonstrate: how the boiling process specifically leads to reduced polyphenol content while increasing acidity and residual sugar; how and why different fermentation temperatures significantly influence the aromatic profile (e.g., complex aromas at lower temperatures vs. flavonoid loss and sulfur compound formation at higher temperatures).
Response 7:Thank you for your suggestion. The conclusion section has been revised.
Through the revisions, the paper has clearly elucidated the mechanistic basis for the reduction in polyphenol content and the concurrent increase in acidity and residual sugar levels resulting from the boiling process, as well as the substantial influence of varying fermentation temperatures on aromatic characteristics.
Comments 8: Please avoid the term "volatile flavor compound". The standard and preferred term in scientific literature is "volatile compound" or "aroma compound". "Flavor" is a multimodal sensation that includes taste (sweet, sour, etc.), which these instruments do not measure.
Response 8:Thank you for your suggestion. The term "volatile flavor compound" in the manuscript has been revised to "volatile compound" throughout.
Comments 9: Ensure strict adherence to the author guidelines of the target journal.
The formatting of references (citation style, journal abbreviations, etc.).
-The correct way to cite literature within the text.
-The structure of headings and subheadings.
-Standardize the scientific style throughout the manuscript and eliminate redundant phrasing.
Response 9: The formatting of references, the correct way to cite literature within the text, and the structure of headings and subheadings have been revised and corrected.
